# Language Model Networks:
# Supervision-Efficient Learning through Dense Communication

**Shiguang Wu** [1]  **Yaqing Wang** [2]  **Quanming Yao** [1 3 4]

## Abstract

Language models are increasingly used not only as standalone predictors but also as components in larger inference systems, from test-time scaling to multi-agent collaboration. We study *language model networks*, where pre-trained language models serve as reusable nodes and intelligence emerges from their topology, communication, and optimization. Existing systems mostly communicate through natural language: easy to deploy, but discrete, inefficient, and hard to optimize from end-task supervision. We propose LMNet, a dense and differentiable realization of this paradigm. LMNet uses stripped LLMs as vertex modules and trainable seq2seq modules as communication edges, enabling intermediate nodes to exchange dense vectors while preserving natural-language input and output at the system boundary. By bypassing intermediate embedding and de-embedding, LMNet enables efficient information transfer, end-to-end gradient optimization, and learned communication beyond hand-designed protocols. Experiments show performance with small additional training cost and effective adaptation under limited supervision.

## 1. Introduction

Large Language Models (LLMs) have achieved impressive performance in language understanding, generation, and reasoning (Brown et al., 2020; Achiam et al., 2023; Yang et al., 2024; Grattafiori et al., 2024), but still struggle on tasks requiring specialized knowledge, complex reasoning, or sustained computation. A growing line of work addresses these

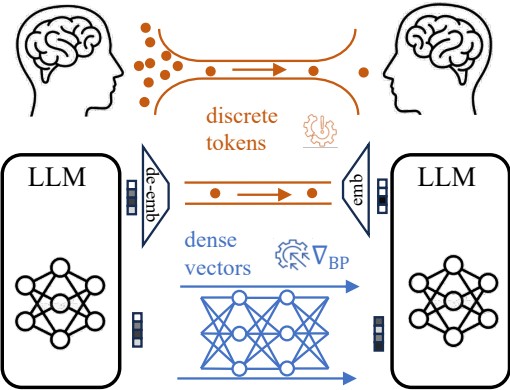

*Figure 1.* Language model networks organize language models as computational nodes connected by communication edges. Natural-language edges are easy to deploy but discrete and hard to optimize; LMNet realizes this paradigm with dense vectors that bypass intermediate embedding/de-embedding and enable end-to-end training.

limitations by designing computational topology and information flow around LLMs. Auto-regressive generation and multi-step reasoning can be viewed as a model communicating with itself (Wei et al., 2022; Muennighoff et al., 2025; Zhang et al., 2025); multi-model and multi-agent systems build explicit networks where multiple LLMs interact with each other and the environment (Hong et al., 2023; Zhou et al., 2025; Zhuge et al., 2024); and self-consistency or debate also rely on structured information exchange (Wang et al., 2022; Du et al., 2023).

This paper studies these systems as *language model networks*: pre-trained language models are reusable computational nodes, and system capability is shaped by topology, communication, and optimization among them. From this perspective, the central question is no longer only how to prompt or fine-tune a single model, but how to organize and optimize a network of model nodes under limited supervision, echoing the broader agenda of data-efficient agentic learning (Wang et al., 2026). This view connects test-time scaling, multi-agent collaboration, workflow optimization, and latent reasoning as different choices of node function, communication medium, and network topology.

The first bottleneck to notice is **communication medium**. Most existing systems use natural language because LLMs are pre-trained on text; this choice is convenient, inter-

[1]Department of Electronic Engineering, Tsinghua University, Beijing, China [2]Beijing Institute of Mathematical Sciences and Applications, Beijing, China [3]Beijing National Research Center for Information Science and Technology, Beijing, China [4]State Key laboratory of Space Network and Communications, Beijing, China. Correspondence to: Quanming Yao <qyaoaa@tsinghua.edu.cn>.

*Proceedings of the $43^{rd}$ International Conference on Machine Learning*, Seoul, South Korea. PMLR 306, 2026. Copyright 2026 by the author(s).

pretable, and API-friendly. However, it also creates a supervision and optimization bottleneck: prompts, reasoning traces, workflows, and protocols are typically specified in human-readable form. Natural language is symbolic and discrete. To be processed by LLMs, tokens are embedded into continuous and dense vectors before model computation and decoded back into tokens at the output (Bengio et al., 2003; Vaswani et al., 2017). Though this embedding/de-embedding cycle is necessary for communication between model and human, it is redundant for communication among models, loses information, and breaks gradient flow. For LLM-to-LLM coordination, we therefore seek to learn communication directly from end-task supervision rather than handcrafting or annotating intermediate natural-language messages.

We propose LMNet, a dense and differentiable realization of language model networks. Instead of relying on discrete tokens to bridge models, one LLM node outputs internal hidden states as dense vectors and another node directly accepts these vectors as input, bypassing intermediate embedding/de-embedding. Under this design, stripped LLMs act as vertex modules, trainable sequence-to-sequence modules act as communication edges. We build the **network topology** which forms a layer-wise fully-connected directed graph analogous to an multi-layer perceptron (MLP), to be task-level-function approximator, while preserving natural-language input and output at the bottom and top ends. Thus, LMNet keeps the external interface of ordinary LLM systems while making internal communication learnable. This shift offers four advantages:

- **Higher information efficiency**: Dense vectors carry richer information per token, reducing loss from embedding and de-embedding.

- **Fully differentiable architecture**: Communication among model nodes becomes end-to-end differentiable, supporting efficient gradient-based optimization.

- **Machine-native communication**: LLM nodes can learn roles and protocols suited for network-level collaboration beyond human-designed natural-language priors, and forming optimized topology.

- **Supervision-efficient coordination**: Communication can be learned from final task supervision without labels for intermediate messages, roles, or reasoning traces.

Our contributions are summarized as follows. First, we formulate *language model networks*, a paradigm that organizes pre-trained language models as computational nodes connected by communication edges. Second, we instantiate this paradigm with LMNet, where stripped LLMs and trainable seq2seq edge modules enable dense, differentiable inter-model communication, under a network topology. Third,

we show that learned communication among LM nodes improves general capability with small additional training cost and supports limited-supervision adaptation. Finally, we analyze the trained network and find that its topology is exploited through learning and that intermediate nodes exhibit nontrivial communication behaviors.

Empirically, we study two applications. For capability improvement, LMNet improves a pre-trained LLM's general reasoning ability using public pre-training datasets and less than 1% of the LLM pre-training cost. For limited-supervision adaptation, LMNet learns communication modules from scarce task data and shows strong performance compared with natural-language communication, latent reasoning methods, and parameter-efficient fine-tuning (PEFT) methods.

## 2. Related Works

We review existing work through the lens of language model networks, where model behavior is improved by adding structure, communication, or optimization around language models. Existing approaches differ mainly in their topology, communication medium, and optimization strategy.

**Multi-step reasoning and test-time scaling.** Multi-step reasoning (Wei et al., 2022; Yao et al., 2023a; Besta et al., 2024; Tian et al., 2024), or more generally test-time scaling (TTS) (Muennighoff et al., 2025), improves LLM performance by allocating additional computation during inference. Chain-of-Thought prompting encourages LLMs to generate intermediate reasoning steps (Wang et al., 2022); self-consistency and best-of-N sampling generate multiple candidate outputs and select among them (Wang et al., 2022); process-based verifiers train models to evaluate intermediate steps and support search over reasoning trajectories (Setlur et al., 2024). These methods can be viewed as simple language model networks in which a model communicates with itself through discrete token sequences.

**Text-based multi-model and agent networks.** Text-based multi-model and agent systems build complex workflows to improve performance by leveraging specialized LLMs to divide tasks into subtasks. Some systems are designed from human prior knowledge, such as standardized operating procedures for software development (Hong et al., 2023) or reasoning-and-acting prompts with tools (Yao et al., 2023b). Others optimize workflows or communication topologies using search (Zhuge et al., 2024; Zhou et al., 2025). These methods are attractive because they can often be built with black-box API calls and natural-language messages. However, their communication remains discrete and non-differentiable, so optimization typically relies on prompt engineering, search, textual feedback, or black-box workflow optimization.

**Latent reasoning and differentiable communication.** Some works study chain-of-thought reasoning in latent space rather than natural language ([Hao et al., 2024](#); [Cheng & Durme, 2024](#); [Shen et al., 2025](#)). These approaches partially move beyond the discrete token bottleneck, but they are largely limited to recurrent inference within a single backbone or simple linear passing. They do not directly construct a higher-level network with complex topologies, heterogeneous model interactions, or modular collaboration. In contrast, LMNet realizes the language-model-network view at the computational level: LLMs are composable nodes in a generalized graph, edges are trainable modules, messages are dense vectors, and communication is learned from final task supervision.

# 3. Language Model Networks

We now present LMNet, a dense and differentiable instance of the broader language model network paradigm. In a language model network, nodes are reusable language models or model-derived modules, edges define how information is communicated, and topology specifies how computation is composed. This paper focuses on a fully differentiable realization: LLMs are treated as vertexes, communication edges are trainable seq2seq modules, and intermediate messages are dense vectors rather than natural-language strings.

## 3.1. Constructing LMNet

The construction includes (i) defining LM nodes by stripping internal embedding/de-embedding layers from LLMs; (ii) introducing trainable edge modules for dense communication between nodes; and (iii) specifying a communication topology over the nodes.

### 3.1.1. LM Nodes: Stripped Transformers

Denote a tokenized text in natural language as $\vec{x}^{\text{in}} = [x_1, x_2, \cdots, x_n]$, a sequence of discrete tokens where each token $x_i \in \mathcal{D}$ and $|\mathcal{D}|$ is the vocabulary size. The embedding layer $E$ embeds the discrete tokens into dense vectors, $\mathbf{X}^{\text{in}} = E(\vec{x}^{\text{in}})$, where $\mathbf{X}^{\text{in}} = [\mathbf{x}_1, \mathbf{x}_2, \cdots, \mathbf{x}_n]$ and $\mathbf{x}_i \in \mathbb{R}^d, d << |\mathcal{D}|$. Inside an LLM, the transformer model $T$ takes $\mathbf{X}^{\text{in}}$ as input, and output dense vectors with the same size, denoted as $\mathbf{X}^{\text{out}} = T(\mathbf{X}^{\text{in}})$. The de-embedding layer $D$ decodes the dense vectors to discrete tokens to output natural language text, denoted as $\vec{x}^{\text{out}} = D(\mathbf{X}^{\text{out}})$. The complete function of an LLM $f$ is $\vec{x}^{\text{out}} = D \circ T \circ E(\vec{x}^{\text{in}}) = f(\vec{x}^{\text{in}})$.

Given two LLMs $f_1, f_2$ with communication flow $f_1 \to f_2$, the existing and natural way is to let them communicate with natural language by setting $\vec{x}_2^{\text{in}} = \vec{x}_1^{\text{out}}$, i.e., $\vec{x}^{\text{out}} = D_2 \circ T_2 \circ E_2 \circ D_1 \circ T_1 \circ E_1(\vec{x}^{\text{in}}) = f_2 \circ f_1(\vec{x}^{\text{in}})$. Note that rather than merely a feeding-forward module, $D$ includes discretization operation like $\arg\max$, which leads

to undesired information loss and cutoff of gradient. Therefore, we propose to remove the internal de-embedding layer $D_1$ and correspondingly removing the embedding layer $E_2$. We let them communicate with dense vectors by setting $\mathbf{X}_2^{\text{in}} = \mathbf{X}_1^{\text{out}}$, i.e., $\vec{x}^{\text{out}} = D_2 \circ T_2 \circ T_1 \circ E_1(\vec{x}^{\text{in}})$. More generally, for communication flow $f_1 \to f_2 \cdots \to f_n$, all internal de-embedding layer and embedding layers will be removed, i.e., $\vec{x}^{\text{out}} = D_n \circ T_n \circ T_{n-1} \circ \cdots \circ T_1 \circ E_1(\vec{x}^{\text{in}})$: there are only one embedding layer $E_1$ and one de-embedding layer $D_n$ to keep natural language input-output of the complete system, while all intermediate LLMs are stripped transformer $T$, communicated through dense vectors. This eliminates internal information loss and enables joint gradient descent for efficient optimization.

### 3.1.2. Communication Edges: Trainable Seq2seq Modules

Another critical component in our method is communication module, or edge module $e$. Each $e$ is a small seq2seq module which will function on a communication path $\mathbf{X}_i^{\text{in}} = e(\mathbf{X}_{i-1}^{\text{out}})$. Note that the stripped transformer $T_i$ from a pre-trained LLM has not learned to take dense vectors $\mathbf{X}_i^{\text{in}} = \mathbf{X}_{i-1}^{\text{out}} \in \mathbb{R}^d$ as input yet, as it is pre-trained with input from a discrete space $\mathbf{X}_i^{\text{in}} = E_i(\vec{x})$ which is very different. Introducing $e$ is not only aimed at aligning them, but also enabling translating and distributing a single $\mathbf{X}_{i-1}^{\text{out}}$ to multiple targets differently with multiple different $e$. This is the key to enable light-weight learning dense communication and complex communication topology.

### 3.1.3. Network Topology

While a language model network can in principle use many topologies, such as chains, trees, sparse graphs, or dynamic routing, we specify LMNet as a layer-wise fully connected feed-forward network for simplicity. This is inspired by the similar topology among neurons in MLP, which enables the universal approximation property ([Hornik et al., 1989](#)). Formally, denote LMNet as $\mathcal{N} = (\mathcal{V}, \mathcal{E})$ where:

- Each vertex $v \in \mathcal{V}$ represents $T$ (a stripped transformer without embedding and de-embedding layer). The vertexes are arranged in $L$ layers: $\mathcal{V} = \{\mathcal{V}^l\}_{l=1}^L = \{\{v_i^l\}_{i=1}^{n_i}\}_{l=1}^L$. Specially, there is only one vertex at the last layer as the output vertex, $n_L = 1$, whose de-embedding layer $D_1^L$ is kept to convert dense vectors to discrete tokens in natural language for final output. And there is only one vertex at the first layer as the input vertex, $n_1 = 1$, whose embedding layer $E_1^1$ are kept to convert discrete tokens in initial input natural language to dense vectors. We denote $v_1^0 = E_1^1$ for notation consistency.

- Each edge $e \in \mathcal{E}$ represents a communication path, through a trainable seq2seq module parameterized by $\omega$.

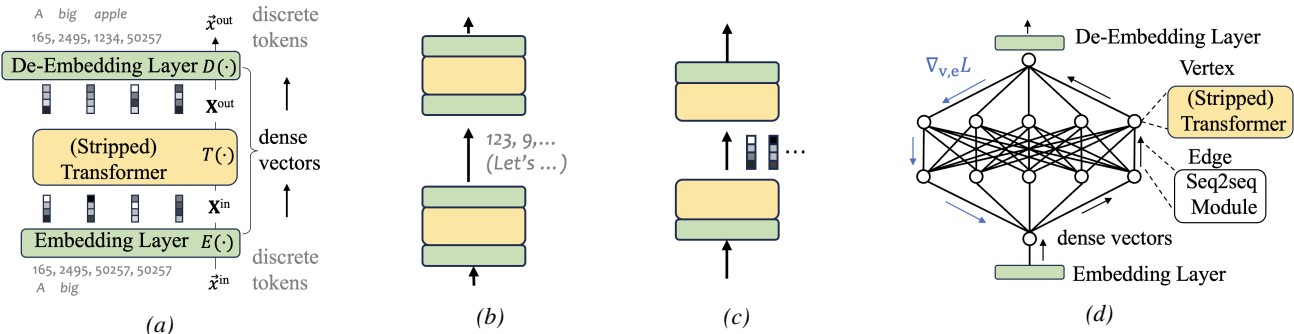

*Figure 2.* Illustration of LMNet as a dense, differentiable realization of language model networks. 2a. A standard LLM embeds discrete token inputs into dense vectors and de-embeds dense outputs back into tokens. 2b. Text-based LM networks communicate through discrete tokens between model nodes. 2c. LMNet bypasses intermediate embedding/de-embedding, enabling direct dense-vector communication. 2d. The resulting network connects stripped transformers with trainable communication modules and can be optimized end-to-end.

As a layer-wise fully connected structure, there are and only are $\{e_{i,j}^l \mid \forall v_i^l \in \mathcal{V}^l, \forall v_j^{l+1} \in \mathcal{V}^{l+1}\}$.

Given initial input text $\vec{x}^{\text{in}}$, the embedding layer $E_1^1$ embeds it to dense vectors $\mathbf{X}^{\text{in}}$. Then, $\mathcal{N}$ works in the way like a feed-forward neural network does. Formally, initializing $\mathbf{X}_1^0 = \mathbf{X}^{\text{in}}$, we have

$$\mathbf{X}_j^{l+1} = v_j^{l+1}\left(\texttt{Agg}(\{e_{ij}^l(\mathbf{X}_i^l; \omega_{ij}^l); \theta_j^{l+1}\}_{v_i^l \in \mathcal{V}^l})\right), \quad (1)$$

where $\theta_j^{l+1}$ is the parameter in vertex module $v_j^{l+1}$ and $\omega_{ij}^l$ is the parameter in edge module $e_{ij}^l$. $\texttt{Agg}$ is aggregation function, where a simple choice we can use is vector summation that $\mathbf{X}_j^{l+1} = v_j^{l+1}\left(\sum_{v_i^l \in \mathcal{V}^l} e_{ij}^l(\mathbf{X}_i^l; \omega_{ij}^l); \theta_j^{l+1}\right)$. It has many other natural and reasonable choices like concatenation. The final output $\mathbf{X}_1^L$ is de-embedded to text $\vec{x}^{\text{out}} = D_1^L(\mathbf{X}_1^L)$, and mapped to the output logits $\boldsymbol{p}^o \in \mathbb{R}^{n \times |\mathcal{D}|}$ for optimization. In this way, such a LMNet takes natural language as input and output in the same way as an LLM system typically does, thus can be applied for general NLP tasks. More details on the specification of the modules (vertex, edge, and aggregation) as well as the topology hypothesis are provided in Section 6.

### 3.2. End-to-End Training

One of the most significant benefits of communication through dense vectors is that the path is differentiable. Given a differentiable supervision signal $\mathcal{L}$, i.e., $\partial\mathcal{L}/\partial\boldsymbol{p}^o$, we can obtain gradient on all parameters in LMNet, i.e., $\partial\mathcal{L}/\partial\theta_i^l$ and $\partial\mathcal{L}/\partial\omega_{ij}^l$ for any $l, i, j$. This enables joint and efficient optimization by end-to-end gradient descent. Importantly, the intermediate dense messages are not directly supervised; they are optimized only through the final supervision signal, allowing LMNet to learn communication protocols without annotations of intermediate messages or roles. Thus, all stages in the pipeline of training a single LLM can be applied to train LMNet, including auto-regressive pre-training,

supervised fine-tuning, training by reinforcement learning. The training data and strategies are to be determined specifically to the application. We do not specify here but will implement exemplar applications in Section 4 by performing end-to-end auto-regressive training.

It is recommended to treat $\boldsymbol{\theta}$ and $\boldsymbol{\omega}$ differently, as $\boldsymbol{\theta}$ has already been pre-trained with fine-grained information, while $\boldsymbol{\omega}$ is randomly initialized that contains no basic knowledge at all. It is recommended to first do pre-training with naive auto-regressive loss optimizing $\boldsymbol{\omega}$ only, to equip LMNet with basic ability to model natural language. The inference strategy is coupled with the training. In this paper we would only implement end-to-end auto-regressive training and decoding for inference. Another natural choice is to make inference by letting each vertex decode to generate complete/multiple-token sequence, aggregating by concatenation, then feeding to latter modules. This can be achieved by implementing such process while still supervise the final output only through post-training. As discussed in Section 6.3.

### 3.3. Reducing Complexity by Vertex Parameter Sharing

It is undeniable that LMNet increases complexity comparing with a single vertex LLM, both in parameter size and inference latency per token. With the proposed parameter-sharing technique, the cost in both aspects may be far less than anticipated. Denote the parameters in LMNet as $\boldsymbol{\theta}$ and $\boldsymbol{\omega}$ as the collection of vertex and edge parameters respectively. Denote the depth of LMNet as $L$ and average width as $W$, approximately with $L \times W^2$ edges following the layer-wise fully-connected topology. Denote the average time-consumption of feeding-forward as $t_\theta$ through a vertex and $t_\omega$ through an edge. With vertex parameter sharing, which means given a single pre-trained LLM $\theta_v$, i.e., initializing all $\theta_i^l$ with $\theta_v$ and keeping $\theta_i^{l_1} = \theta_j^{l_2}$ for any $l_1, l_2, i, j$ during training, the parameter size is $|\boldsymbol{\theta}| + |\boldsymbol{\omega}| = |\theta_v| + L \times W^2 \times |\omega|$ where $|\omega| \ll |\theta_v|$. This means when the size of LM-

Net grows, its parameter size only grows by a small ratio. This gives the possibility to build a deep and wide LM-Net. For time complexity, keeping vertexes in the same layer identical enables simply paralleling them through data-parallel. In this case, a feeding-forward process only requires a sequential processing with length $L$, like a MLP, and $t = L \times t_\theta + L \times W^2 \times t_\omega$, where $t_\omega \ll t_\theta$.

# 4. Experiments

We evaluate whether learning communication in a language model network can improve intelligence under limited additional supervision. Specifically, we ask two questions. First, can a network of reusable LM nodes improve general capability with small additional training cost? Second, can learned communication edges adapt a pre-trained LM to downstream tasks when supervision is scarce? These two settings instantiate the same principle: instead of supervising intermediate messages, roles, or reasoning traces, LM-Net learns how information should flow among model nodes from final task supervision. Code is provided at https://github.com/LARS-research/LMNet/.

## 4.1. Learning Communication for General Capability

Society of Mind theory (Minsky, 1986) suggests that higher-level intelligence emerges from the coordination of simpler components. LMNet embodies this idea as a language model network that coordinates reusable pre-trained LM nodes: each vertex (a stripped transformer from pre-trained LLM) acts as a modular component, while the edges (trainable seq2seq modules) facilitate differentiable communication, enabling the system to collectively solve problems beyond the reach of any single LLM. By optimizing these communications, LMNet can improve not only computational scale but also the way information is routed and transformed across model nodes.

### 4.1.1. Settings

Due to the computation budget, we consider modern LLMs with minimum size. We use Qwen2.5-0.5B as vertex module to be shared among all vertexes. We implement LMNet with 5 layers with 1/4/4/4/1 vertexes in each layer. We use an attention block with the same structure as one transformer layer in the vertex module for each edge module. These edge modules are independently random initialized and to be optimized. Such a LMNet has 1.1B parameters in total. All data used in this study come from public datasets without testing leakage. The details are provided in Appendix A. As mentioned above, we first freeze vertex parameters and only update edge parameters with the typical auto-regressive loss, then we update all parameters together.

We compare the trained LMNet with other solutions based on Qwen2.5-0.5B, including **Prompt** (using ad-hoc prompt for best performance), and **SFT** (using the same training data as LMNet to fine-tune all parameters of a Qwen2.5-0.5B). We also compare with representative TTS methods Self-Refine (Madaan et al., 2023) and Self-Consistency (Wang et al., 2022), with similar test-time computation budget as LMNet. We evaluate the trained LMNet on widely-used benchmarks, with details provided in Appendix A.

### 4.1.2. Performance

The performance results are provided in Table 1. First, LM-Net brings consistent and significant improvement compared with Prompt (+30.5%), which can be viewed as natural-language communication with the model itself. This indicates the advantage of the proposed dense communication medium and topology, and gives a way to improve a pre-trained LLM using public data and acceptable additional training cost.

Second, one may argue that the improvement mainly relies on training data. SFT (implemented by very moderate training configurations) can hardly bring improvement over the pre-trained LLM (-5.7%). In fact, these public data is likely to be seen during the pre-training of LLM. This indicates the benefit of LMNet comes from the learned communication, rather than the training data.

Third, one may argue that similar improvement could be achieved by increasing test-time computation. We compare LMNet with representative parallel TTS strategy Self-Consistency and sequential TTS strategy Self-Refine. As each feed-forward process of LMNet contains 1+4+4+4+1=14 repeats of Qwen2.5-0.5B and additional 0.6B edge modules (no repeat), we compare with above TTS with a budget of 16 trials/iterations ($\times 16$). The results show that LMNet significantly outperforms both methods (+30.5% vs. +0.5%/+3.4%). Another advantage of LMNet over TTS methods is that it is generally applicable. For example we could not apply Self-Consistency on coding tasks without external feedback as the consistency of code is not defined, let alone other TTS methods requiring additional verifier or step-wise decomposition.

Finally, one may argue that the improvement comes from additional parameters, so we should compare LMNet (1.1B) with other pre-trained LLMs with similar sizes. This would not be a fair comparison, because LMNet is a general method to improve intelligence of a given LLM, with additional training cost only less than $1\%$ of the pre-training. Note that this paper aims to provide a method rather than the trained LMNet ready-to-use. Still, we compare with modern pre-trained LLMs in similar size Appendix A, which shows better/comparable performance with models in similar/larger size. We also discuss the opportunities of LM-Net as a method for scaling for general intelligence, which

*Table 1.* Performance comparison on Qwen2.5-0.5B with different methods. (Accuracy, %). N.a. means not applicable.

| | Tasks | Prompt | SFT | Self-Refine (×16) | Self-Consistency (×16) | **LMNet** |
|---|---|---|---|---|---|---|
| General | MMLU | 44.3 | 44.6 | 45.2 | 46.0 | **53.9** |
| | MMLU-Pro | 15.7 | 13.2 | 16.4 | 17.1 | **26.2** |
| | BBH | 20.3 | 19.9 | 24.7 | 21.6 | **47.3** |
| | ARC-C | 35.6 | 34.6 | 32.8 | 36.3 | **38.0** |
| | TruthfulQA | 40.2 | 40.5 | 38.9 | 41.3 | **47.9** |
| Math & Science | GSM8K | 41.6 | 41.8 | 42.1 | 43.5 | **50.3** |
| | MATH | 19.5 | 12.4 | 19.8 | 19.0 | **38.8** |
| | GPQA | 24.8 | 22.0 | 23.5 | 25.2 | **25.6** |
| | MMLU-stem | 39.8 | 39.8 | 40.6 | 41.4 | **46.0** |
| Coding | HumanEval | 30.5 | 27.6 | 28.7 | N.a. | **39.0** |
| | MBPP | 39.3 | 35.1 | 40.5 | N.a. | **45.8** |
| Relative Improvement | | 0.0% | -5.7% | +0.5% | +3.4% | **+30.5%** |

expects advantage in training cost and data consumption comparing with training a monolithic LLM from scratch.

#### 4.1.3. ANALYSIS OF LEARNED COMMUNICATION

We take a closer look at the trained LMNet to see whether supervision through final outputs induces meaningful communication among LM nodes, rather than merely adding parameters or computation. The results and detail analysis is provided in Appendix B. We reach conclusions that the layer-wise fully connected structure is fully exploited through training, without distinguishable substructures, and different vertexes perform different behaviors.

### 4.2. Supervision-Efficient Adaptation with Limited Data

Now we discuss another scenario where LMNet can be effectively applied: adapting pre-trained LLMs under limited supervision. We consider a strict setting where only a pre-trained LLM and a task training set are accessible, and the goal is to improve performance on an unseen test set from the same domain. Under this setting, the common practice is parameter-efficient fine-tuning (Houlsby et al., 2019; Li & Liang, 2021; Hu et al., 2022), which avoids overfitting when data are scarce. LMNet offers another option: build a collective intelligence system and let the limited task supervision determine the dense communications. This can be done by constructing a LMNet with shared vertex parameters as regularization, and/or freezing pre-trained vertexes while training only edge parameters. No supervision on intermediate messages, roles, or reasoning processes is required.

Generally, we consider three categories of methods: (i) prompting, including demonstrating examples from the training set for in-context learning (ICL) and prompting the model to reason better, e.g., with CoT; (ii) fine-tuning, including latent reasoning and parameter-efficient fine-tuning methods; and (iii) training LMNet. We consider MMLU,

GSM8K, and E2E (Novikova et al., 2017) respectively.

#### 4.2.1. IMPROVING REASONING ABILITY ON MMLU AND GSM8K

We study on widely used benchmark MMLU and GSM8K to show the effect of LMNet on different models. The baselines include **LMNet**: construct LMNet with the given LLM as vertex and train with training set. **Pred**: directly ask the LLM to answer the question; **Prompt**: use model- and dataset- specific prompt engineering to optimize the performance (typically and at least combines ICL and CoT); **SFT**: fine-tune the LLM with the training set, including learning for latent reasoning. Specifically, as these two datasets focus on the reasoning ability, there are two additional baselines which do latent reasoning, which can be viewed as learning communication in latent space rather than natural language: **COCONUT** (Hao et al., 2024) and **CODI**(Shen et al., 2025). The difference between LMNet between them is LMNet enabling complex topology and trained end-to-end auto-regressively, while they only communicate iteratively with the LLM itself, trained by un-supervising some tokens or self-distillation.

For both datasets, we construct LMNet with a small scale (3 layers with 1/2/1 vertexes), and keep all the vertexes share the same group of parameters for efficiency. All parameters in LMNet are updated together by gradient descent. We use a module with the same structure with a single transformer layer from the corresponding backbone LLM. Taking Qwen2.5-1.5B for example, a single LLM model has $1.76 \times 10^9$ parameters, while constructing the LMNet only introduces additional $2.45 \times 10^8$ parameters on the edge modules. In this case, the parameter size of LMNet is close to a single model, and the cost of training LMNet and fine-tuning a single model have similar scale. The performance on MMLU is evaluated by $\Delta$Acc, which is the difference between testing accuracy and random guess average accuracy (25%), provided in Table 2. For GSM8K,

*Table 2.* Performance comparison on MMLU dataset ($\Delta$Acc, %).

|  | GPT2-XL | Llama3.2-1B | Llama3.2-1B-Instruct | Qwen2.5-0.5B | Qwen2.5-1.5B |
|---|---|---|---|---|---|
| Pred | 0.24 | 6.05 | 20.93 | 22.36 | 34.75 |
| Prompt | 1.69 | 11.69 | 24.30 | 22.50 | 35.83 |
| SFT | 7.40 | 24.65 | 24.83 | 21.50 | 35.02 |
| COCONUT | 3.63 | 9.88 | 18.54 | 19.22 | 33.85 |
| CODI | 5.01 | 20.69 | 21.30 | 19.48 | 35.16 |
| **LMNet** | **13.10** | **27.19** | **27.57** | **23.35** | **37.28** |

*Table 3.* Performance comparison on GSM8K dataset (Accuracy, %).

|  | Llama3.2-1B | Llama3.2-1B-Instruct | Qwen2.5-0.5B | Qwen2.5-1.5B |
|---|---|---|---|---|
| Pred | 2.88 | 30.10 | 5.31 | 9.25 |
| Prompt | 11.49 | 43.67 | 41.60 | 68.50 |
| SFT (w/ CoT) | 38.69 | 46.39 | 42.94 | 69.31 |
| COCONUT | 26.79 | 38.27 | 30.81 | 48.63 |
| CODI | 37.25 | 44.58 | 42.06 | 65.30 |
| **LMNet** | **44.02** | **56.15** | **50.82** | **72.67** |

we noticed that the training set of GSM8K is much smaller than MMLU (7.47k vs 100k), which seriously constrains the methods' effect, as reported in Appendix C. So we add additional 93.7k training data from OpenR1-Math-220k[1] for all baselines requiring training. Results are provided in Table 8.

LMNet shows a consistent performance advantage over the other baselines across considered LLMs on both benchmarks. Compared with COCONUT and CODI, which can also be viewed as dense-space communication methods, LMNet is not restricted to a chain-like reasoning structure and does not supervise intermediate communication to mimic explicit CoT. Instead, its fully connected topology lets the model learn a richer communication pattern from final task supervision, contributing to the performance advantage.

#### 4.2.2. DATA- AND SUPERVISION-EFFICIENT ADAPTATION ON E2E DATASET

A common issue in adapting/fine-tuning LLMs is data scarcity, due to the mismatch between limited task data and massive parameter amount. Such issue is typically addressed through PEFT methods, which identify a small set of parameters for effective adaptation while avoiding over-fitting. LMNet gives another option: learning data-dependent dense communication while reusing the pre-trained vertex model. In this part, we compare with PEFT methods following the experiment setting in LoRA (Hu et al., 2022): we study GPT2-M (Radford et al., 2019) on the E2E dataset, which is known to require effective adapta-

tion with limited data, and consider the same group of PEFT methods (Houlsby et al., 2019; Li & Liang, 2021; Hu et al., 2022).

To avoid over-fitting, by default we only train the edge parameters in LMNet and keep the vertex parameters frozen. We still construct LMNet with a small scale (3 layers with 1/2/1 vertexes), using an attention block containing 5.25M parameters for each edge module. The results are provided in Table 4. LMNet performs best, showing that learning communication edges gives effective and generalizable adaptation under limited supervision. Thanks to the fully differentiable paths in LMNet, we can also integrate LMNet with PEFT methods. Results in Appendix D suggest that plugging in appropriate PEFT methods can further improve the performance.

## 5. Ablation Studies

We study the effect of LMNet on larger vertex LMs, and with different architecture (depth/width). Due to each case requires training a LMNet independently, the cost of training multiple LMNet on large-scale general data is unacceptable for us. We could only perform ablation study about LMNet on customizing LLMs with limited data.

**Different Vertex Model Size** We have the following results using Qwen2.5-3B and 7B models following the same setting in Section 4.2.1. Table 5 shows the results, where LMNet shows consistent advantage.

**Different LMNet Depth/Width** We follow the same setting in Section 4.2.2, using GPT2-M on E2E dataset. Table 6 shows the results. While the width/depth range is limited,

---

[1]`https://huggingface.co/datasets/open-r1/OpenR1-Math-220k`

*Table 4.* Performance comparison on E2E dataset with GPT2-M. * indicates results from (Hu et al., 2022).

| Method | Metrics ↑ | | | | | Rank Avg. |
|---|---|---|---|---|---|---|
| | BLEU | NIST | MET | ROUGE-L | CIDEr | |
| No Adaptation | 0.00 | 0.42 | 0.04 | 0.16 | 0.00 | 9.0 |
| FT* | 68.2 | 8.62 | 46.2 | 71.0 | 2.47 | 4.5 |
| Adapter$^L$(0.37M)* (Lin et al., 2020) | 66.3 | 8.41 | 45.0 | 69.8 | 2.40 | 8.0 |
| Adapter$^L$(11.09M)* (Lin et al., 2020) | 68.9 | 8.71 | 46.1 | 71.3 | 2.47 | 3.9 |
| Adapter$^H$* (Houlsby et al., 2019) | 67.3 | 8.50 | 46.0 | 70.7 | 2.44 | 6.7 |
| FT$^{Top2}$* (Li & Liang, 2021) | 68.1 | 8.59 | 46.0 | 70.8 | 2.41 | 6.3 |
| PreLayer* (Li & Liang, 2021) | 69.7 | 8.81 | 46.1 | 71.4 | 2.49 | 2.7 |
| LoRA (Hu et al., 2022) | 68.9 | 8.68 | **46.5** | **71.5** | **2.51** | 2.3 |
| **LMNet** | **70.5** | **8.85** | **46.5** | **71.5** | 2.48 | **1.6** |

*Table 5.* Performance (Accuracy, %) with different model sizes.

| Data | Model Size | Prompt | SFT | **LMNet** |
|---|---|---|---|---|
| MMLU | 3B | 65.6 | 64.5 | **67.2** |
| | 7B | 74.2 | 73.8 | **76.0** |
| GSM8K | 3B | 79.1 | 80.3 | **81.9** |
| | 7B | 85.4 | 85.5 | **87.1** |

no significant difference or patterns can be observed.

*Table 6.* Performance with different LMNet architecture.

| LMNet Arc. | Metrics ↑ | | | | |
|---|---|---|---|---|---|
| | BLEU | NIST | MET | ROUGE-L | CIDEr |
| 1/1 | 69.1 | 8.76 | **46.6** | 70.6 | 2.43 |
| 1/2/1 | **70.5** | **8.85** | 46.5 | **71.5** | 2.48 |
| 1/2/2/1 | 69.8 | 8.80 | 46.0 | 71.1 | **2.49** |
| 1/4/1 | 69.1 | 8.70 | 45.9 | 70.9 | 2.44 |
| 1/4/4/1 | 68.7 | 8.79 | 46.5 | 71.2 | 2.48 |

# 6. Discussion

## 6.1. Specification of Modules

For edge modules, any seq2seq modules are applicable. Note that this can be reduced to element-wise functions like a MLP processing each vector independently. To be parameter-efficient yet expressive enough, we choose to use a single attention-block (1-layer transformer) for each edge. All edge modules would be independently and randomly initialized. For vertex models, LMNet does not require all the vertex models to be identical. We can implement each vertex with different pre-trained LLM respectively, to exploit that different LLMs may have different expertise and LMNet can combine them together. However, we can choose a much more parameter-efficient way: implementing all the vertex with a single pre-trained LLM. As different edges are different, the input information of different vertex would be different, so LMNet would still be effective. For

the aggregation function of messages from multiple input edges of one vertex, we use sum for simplicity: in this case we can use the same causal mask for all vertexes and edges, to keep the causal structure of the input sequence of pre-trained LLMs.

Another consideration we have is that we might not use 'sum' as aggregation, because it can result in information loss. We can use 'concatenate' for aggregation, like reading language. As the topology of LMNet architecture is pre-defined, this can be implementable by setting the order among vertexes in the same layer, and scheduling output length in different layers.

## 6.2. Topology Hypothesis

We construct the layer-wise fully connected structure as the hypothesis of workflow of an LLM system by default. This is inspired by the similar topology among neurons in MLP. Akin to the universal approximation property of MLPs, LMNet can express a wide range topology and functions of workflows. For example, one can easily figure out chain/tree/graph structures inside the fully-connected structure. There are also many other reasonable topology choices, like several paths with no intermediate intersection, skip or residual connection, or mimicking message passing on given graph. One can design the topology according to problem-specific prior knowledge, expertise in vertexes, and expressiveness theory.

## 6.3. Alternative Training Strategy

The end-to-end auto-regressive implementation does not exactly match the image of replacing inefficient NL messages in LLM system. The current performance gain should be attributed to more test-time computation, where each vertex has been equipped with some newly assigned function through training, rather than the imaged sentence-by-sentence communication manner. This is consistent with our observation in Appendix B.2, that no semantically readable sentence can be found if de-embed intermediate outputs fol-

lowing the end-to-end auto-regressive manner, while some vertex outputs can still retain language-like structure due to parameter sharing with the final output vertex.

A closer analogue to multi-step message passing would allow internal vertexes to communicate sentence-level message through generating dense sequences in a inner-auto-regressive rather than end-to-end auto-regressive way. A previous vertex would generate a complete sequence of dense vectors auto-regressively to itself, and then feeding-forward to communication. Each internal vertex could produce a latent sequence for later vertexes, and messages from multiple predecessors could be combined by summation, concatenation, or learned routing. And output of later vertexes would never input to previous vertexes. The final vertex would remain the only component required to generate natural language, while optional supervision on selected intermediate vertexes could be added when oracle reasoning steps are available. Such variants introduce a trade-off: keeping intermediate messages latent preserves supervision efficiency and machine-native communication, whereas intermediate textual supervision may improve transparency and regularization.

## 7. Conclusion, Limitations, and Future Work

In this paper, we introduced LMNet as a dense, differentiable realization of language model networks. Rather than viewing dense communication as an isolated mechanism, LMNet starts from a broader premise: pre-trained language models can be reused as computational nodes, and intelligence can be improved by learning how information flows among them. By stripping intermediate embedding and de-embedding layers, placing LLMs as vertexes in a directed graph, and connecting them with trainable seq2seq edge modules, LMNet enables end-to-end optimization of inter-model communication from final task supervision. Experiments show that this learned communication improves LLM general capability with small additional training cost and supports adaptation under limited supervision.

**Limitations.** The most noteworthy limitation of LMNet is its complexity. As discussed in Section 3.3, LMNet increases parameter size and inference latency compared with a single vertex LLM. Vertex parameter sharing and lightweight edge modules reduce this cost, and the empirical results show advantages over both the vertex LLM and monolithic LLMs with similar size, but dense language model networks are still more expensive than prompt-only or API-based natural-language agent systems. Another limitation is access: dense communication requires access to model internals and parameter optimization, whereas text-based language model networks can be built with black-box APIs. Dense messages are also less directly interpretable

than natural-language messages, creating a trade-off between differentiable optimization and human readability. Finally, as discussed in Section 6.3, the end-to-end auto-regressive decoding studied in this paper does not yet realize sentence-level message passing in an LLM system. Therefore, the current LMNet should be viewed as a dense differentiable network of LM modules rather than a full replacement of sentence-by-sentence message passing among language agents. This limits its ability to model internal multi-step generation, variable-length latent messages, and more explicit communication dynamics among vertexes.

**Future Work.** The most important direction for future work is to implement inner-auto-regressive LMNet, where intermediate vertexes can auto-regressively generate dense latent sequences and pass them to later vertexes before the final natural-language decoding. Such a design would better match the motivating picture of replacing natural-language inter-agent messages with machine-native dense messages, while preserving end-to-end supervision from the final task output. Beyond this direction, language model networks define a broader research agenda. Different instantiations may use natural-language messages, API-based agents, dense vectors, latent states, tool calls, memory modules, or hybrid communication edges. Natural-language and API-based networks are likely to enable rapid near-term applications because they require no model-parameter access, while dense differentiable networks provide a more principled route for learning communication from supervision. Future work also includes richer module specifications, more expressive topology hypotheses, dynamic routing, multiple input/output vertexes for environment interaction, and hybrid supervision strategies that balance latent communication with interpretability. More broadly, we view LMNet as a step toward supervision-efficient intelligence systems where models, agents, memories, tools, and feedback modules are organized as learnable computational networks.

## Impact Statement

This paper presents work whose goal is to advance the field of machine learning. There are many potential societal consequences of our work, none of which we feel must be specifically highlighted here.

## Acknowledgment

Q. Yao is supported by Beijing Science and Technology Program (No.Z251100008125003) and Beijing Natural Science Foundation (No.F251001). Y. Wang is sponsored by Beijing Nova Program.

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

# A. Comparing LMNet with Monolithic LLMs in Similar Size

All data used in this study come from public datasets: C4 (Raffel et al., 2020), Alpaca (Taori et al., 2023), ProsocialDialog (Kim et al., 2022), LaMini-instruction (Wu et al., 2024), MMLU (Hendrycks et al., 2021a) (auxiliary_training split only), MATH (Hendrycks et al., 2021b) (training split only), GSM8K (Cobbe et al., 2021) (training split only), MBPP (Austin et al., 2021) (training split only). Note that due to computation budget, we report the performance of LMNet trained for less than 200 GPU·days (NVIDIA A100), and less than 0.1T tokens or 2e6 PFLOPs in total. This means only a small subset of above mentioned datasets are used.

## A.1. Comparing with Pre-Trained LLMs

*Table 7.* Performance comparison of pre-trained LLMs (Accuracy, %).

| Model | | Qwen2.5-0.5B | **LMNet-1B** | Llama3.2-1B | Qwen2.5-1.5B | Gemma2-2B | Llama3.2-3B |
|---|---|---|---|---|---|---|---|
| # Parameters | | 0.49B | 1.14B | 1.23B | 1.54B | 2.61B | 3.21B |
| # Training Tokens | | 18T | +0.1T | 15T | 18T | 15T | 15T |
| General Tasks | MMLU | 44.3 | 53.9 | 32.2 | **60.9** | 52.2 | 58.0 |
| | MMLU-pro | 15.7 | 26.2 | 12.0 | **28.5** | 23.0 | 22.2 |
| | BBH | 20.3 | **47.3** | 31.6 | 45.1 | 41.9 | 46.8 |
| | ARC-C | 35.6 | 38.0 | 32.8 | 54.7 | 55.7 | **69.1** |
| | Truthfulqa | 40.2 | **47.9** | 37.7 | 46.6 | 36.2 | 39.3 |
| Math & Science | GSM8K | 41.6 | 50.3 | 9.2 | **68.5** | 30.3 | 12.6 |
| | MATH | 19.5 | **38.8** | - | 35.0 | 18.3 | - |
| | GPQA | 24.8 | **25.6** | 7.6 | 24.2 | 25.3 | 6.9 |
| | MMLU-stem | 39.8 | 46.0 | 28.5 | **54.8** | 45.8 | 47.7 |
| Coding | HumanEval | 30.5 | **39.0** | - | 37.2 | 19.5 | - |
| | MBPP | 39.3 | 45.8 | - | **60.2** | 42.1 | - |

We compare the trained LMNet in Section 4.1 (shared Qwen2.5-0.5B as vertexes, 1/4/4/4/1 structure. 1.1B parameters in total) with modern pre-trained-only LLMs in similar size, including Qwen-0.5B/1.5B (Yang et al., 2024), Llama3.2-1B/3B (Grattafiori et al., 2024), Gemma2-2B (Team et al., 2024), on widely-used benchmarks (with commonly-used benchmark-specific prompt) MMLU (Hendrycks et al., 2021a) (5-shot), MMLU-Pro (Wang et al., 2024) (5-shot, CoT), BBH (Suzgun et al., 2022) (3-shot, CoT), GSM8K (Cobbe et al., 2021) (4/8-shot, CoT), MATH (Hendrycks et al., 2021b) (0/4-shot, CoT), GPQA (Rein et al., 2024) (5-shot, CoT), HumanEval (Chen et al., 2021) (0-shot), MBPP (Austin et al., 2021) (0-shot). The performance is provided in Table 7. First, LMNet-1B brings comprehensive and significant improvement over the vertex model Qwen2.5-0.5B. This gives a way to improve general performance given a pre-trained LLM, using public data and acceptable training cost. Second, comparing LMNet-1B with other open-source pre-trained LLMs with similar or slightly larger size, LMNet-1B shows comparable or even better performance. This gives an efficient and effective way to scale for general intelligence by utilizing existing pre-trained LLMs rather than train single LLM from scratch.

Note that we aim at providing a new method rather than the trained LMNet-1B ready-to-use. As we had very limited computation budget, we expect further improvement, by larger-scale training with more deliberate data and schedule, deeper and wider LMNet structure, larger and more diverse vertex modules, and integrating post-training and reinforcement learning.

## A.2. Discussion about Scaling for General Intelligence

The pursuit of general intelligence through LLMs has traditionally focused on scaling individual models (Kaplan et al., 2020; Brown et al., 2020; Achiam et al., 2023; Grattafiori et al., 2024; Yang et al., 2024), either by increasing their parameter count or enhancing their training data. However, this approach faces diminishing returns due to computational costs of the challenges of further optimizing monolithic architectures, and the gradual consumption of high-quality data. Our proposed LMNet paradigm offers a alternative by leveraging existing pre-trained LLMs as modular components within a densely

connected network.

From a technical perspective, comparing with training a single LLM, LMNet has advantage in training cost and data requirement. Note that people need to train from scratch to build a larger LLM, and keep requiring new high-quality data to enhance performance. The first problem is training from scratch is wasting existing pre-trained LLMs which have already encoded vast information. LMNet addresses this by utilizing pre-trained LLMs in vertexes. The second problem is high-quality data is gradually depleted. LMNet addressed this by that LMNet can be trained with the data that has been used for training vertex LLMs, to enable the communication path and adapt transformer to dense vector inputs. Note that existing collection of LLMs could not be optimized for general intelligence effectively, because they communicate through natural language, disabling large-scale efficient optimization through gradient-descent.

From a motivational perspective, by taking pre-trained LLMs as relative lower-level intelligence, LMNet embodies the collective intelligence, the important idea that has been widely claimed and practiced. In LMNet, each vertex (a stripped transformer from pre-trained LLM) acts as a modular component, while the edges (trainable seq2seq modules) facilitate differentiable communication, enabling the system to collectively solve problems beyond the reach of any single LLM. By optimizing these communications, it can be expected that LMNet not only scales computational power but also fosters emergent behaviors, such as advanced reasoning and adaptive problem-solving, that transcend the capabilities of its individual components. Note that with more computation budget, one can also increase the intelligence versatility by implementing different vertexes without parameter-sharing, with LLMs from different expertise/families/sizes.

## B. Analysis of Learned Communication in LMNet

We take a closer look at the trained LMNet, to see what communication has been learned that improves LLM's general intelligence de facto.

### B.1. On Macro-Level

First, we inspect topology patterns across the edges by visualizing and analyzing the parameters in all edge modules of the edges connecting the 1/4/4/4/1 vertexes. We reach a conclusion that the layer-wise fully connected structure is fully exploited through training, without distinguishable substructures.

We try to find some topology patterns across the edges. We pick out the parameters in all edge modules of the edges connecting the 1/4/4/4/1 vertexes, for visualization and analysis (query/key/value/output projection matrix respectively). As provided in Figure 456 and 7, they show similar statistical patterns: though they are different and from each other and initialization, it is hard to distinguish which edge is more significant. We can draw the following conclusions:

- All edges are obviously different from each other, and statistically different from the random initialization.

- No significant pattern among the edges, of statistical information (including std, max/min singular value) of each edge's parameters, can be witnessed.

- Above results indicate that the layer-wise fully connected structure is fully exploited through training, that it does not collapse to a simpler structure, e.g., chains or trees.

Note that above results do not indicate the layer-wise fully connected structure is necessary or optimal. A LMNet with other topology hypothesis could show similar results.

### B.2. On Micro-Level

Second, we perform case study to see the inference process. The case is the first test case in GSM8K, with input text shown in blue block in Figure 3. The trained LMNet answers correctly with output text shown in the green block.

To see if the communication makes effect, we first try to de-embed the input dense vector sequences of different intermediate vertexes, i.e., the output vectors of edge modules. But we failed because most tokens typically show very close logits on many words, and even if we force to find a word with hard-max logit, the resulting input text is not even English (e.g., the red block in Figure 3). However, we find the output vectors of intermediate vertexes, i.e., the input vectors of edge modules, can be de-embedded to meaningful sentences. As shown in the lower part of Figure 3, we de-embed the output sentences

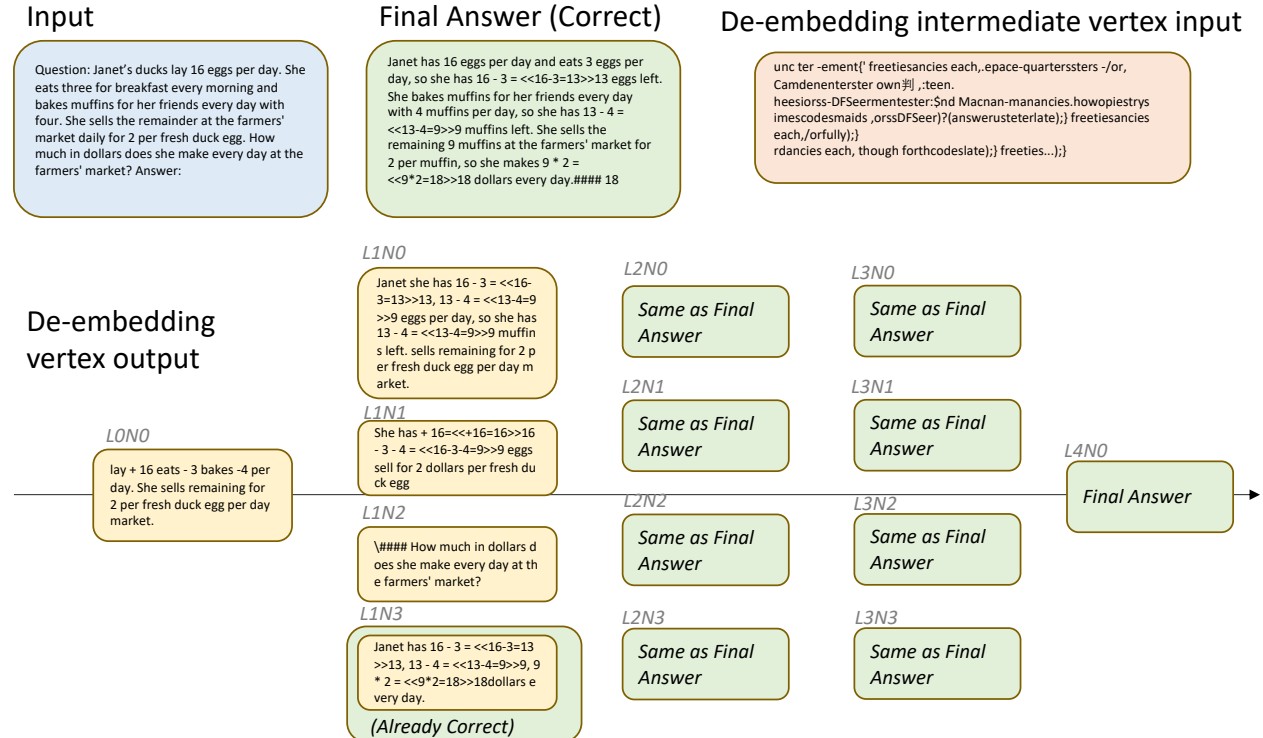

*Figure 3.* Case study of learned communication in trained LMNet.

of all the 1/4/4/4/1 vertexes and truncate to visualize. To obtain the output of certain intermediate vertex, we feed the the intermediate output back to the input vertex auto-regressively, i.e., cutting off all vertexes at latter layers and the other vertexes at the same layer, while keeping all former modules. We de-embed the output sentences of all the 1/4/4/4/1 vertexes and process visualize: truncation begins at the position at input length and ends at first ¡eos¿ token; escape characters are removed for readability. We have the following observation and conclusions:

- The trained LMNet makes inference generally in the expected "division/multi-step reasoning" way through communication. Correct and confident answer can be achieved by early vertexes for easy problem. In this case, the correct answer "18" is first output by L1N3, and all later vertexes output the same sentence as the final output.

- Intermediate vertexes, especially earlier vertexes, no longer necessarily output complete answer or coherent sentence. They tend to aggregate important information at very early positions in the sentence, thus output shorter sequences compared with the output vertex. This is because of the end-to-end auto-regressive manner. This lets the final output condition on compact information produced by intermediate vertexes.

- The input space of the intermediate vertexes departs far from the word embeddings, but the output space of them remains similar. This input-space departure can be interpreted by the fact that we implement the topology with layer-wise fully connected and the aggregation function of different input edges on a single vertex with simple summation, which heavily fuses information; and the joint optimization of vertex, edges, embedding parameters. The output spaces are similar can be interpreted as the parameter-sharing of vertex modules: the LLMs used as intermediate vertexes share parameters with the final output vertex, which are supervised to generate natural language by the word embeddings.

## C. Customizing with Varying Data, and on Challenging Benchmarks

The training set of GSM8K is very small (7.47k sentence). If trained on such a small training set only, it will seriously constrains the effect of methods that updating model parameters (including LMNet and SFT). We provide the performance under such circumstance in Table 9. Given Llama3.2-1B or Llama3.2-1B-Instruction as backbone LLM, LMNet still performs the best. However, given Qwen2.5-0.5B or Qwen2.5-1.5b, LMNet fails in comparison with Prompt, and the

performance of Prompt and LMNet on Llama3.2-1B-Instruction are close. We infer the following two factors together cause such result. First, the training set of GSM8K is much smaller than MMLU (7.47k sentences vs 100k sentences). This result in overfitting risk for methods that updates model parameter, including SFT and LMNet. Despite of this factor, comparing with SFT, LMNet shows advantage, summarized as Table 8. So we have report add additional training data in the experiments reported in main text (Table 3).

Second, these two LLMs are strong enough for such tasks with a single model given proper instructions, that the additional communication steps brought by LMNet do little help. This can also be witnessed in Table 2, from the relative close performance between different methods on these three LLMs comparing with the other weak LLMs. So we further make verification that LMNet would show more advantage on more challenging benchmarks. We test the performance on OpenR1-Math-220k (an unseen subset, much more challenging than GSM8K). Table 10 shows the results. We find that the performance gain of LMNet is much significant comparing with easy tasks and comparing with other baselines, which indicates that LMNet would have larger effect on more challenging tasks, with more necessity of communication.

*Table 8.* Performance comparison of varying training data tested on GSM8K dataset (Accuracy, %).

|  | Qwen2.5-0.5B | Qwen2.5-1.5B |
| --- | --- | --- |
| Prompt | 41.6 | 68.5 |
| SFT (7.47k training) | 24.1 | 51.1 |
| LMNet (7.47k training) | 30.9 | 60.0 |
| SFT (7.47k+93.7k training) | 42.9 | 69.3 |
| LMNet (7.47k+93.7k training) | 50.8 | 72.6 |

*Table 9.* Performance comparison on GSM8K dataset (Accuracy, %).

|  | Llama3.2-1B | Llama3.2-1B-Instruct | Qwen2.5-0.5B | Qwen2.5-1.5B |
| --- | --- | --- | --- | --- |
| Pred | 2.88 | 30.10 | 5.31 | 9.25 |
| Prompt | 11.49 | 43.67 | **41.60** | **68.50** |
| SFT | 25.32 | 35.63 | 24.11 | 51.08 |
| LMNet | **33.41** | **45.75** | 30.93 | 60.02 |

*Table 10.* Performance comparison tested on OpenR1-Math-220k dataset (Accuracy, %).

|  | Qwen2.5-0.5B | Qwen2.5-1.5B |
| --- | --- | --- |
| Prompt | 18.0 | 29.0 |
| SFT (7.47k+93.7k training) | 23.2 | 34.7 |
| LMNet (7.47k+93.7k training) | 29.0 | 46.0 |

## D. Integrating LMNet with PEFT Methods

*Table 11.* Performance comparison on E2E dataset with GPT2-M. * indicates results from (Hu et al., 2022).

| Method | Metrics ↑ | | | | | Rank Avg. |
| --- | --- | --- | --- | --- | --- | --- |
|  | BLEU | NIST | MET | ROUGE-L | CIDEr |  |
| No Adaptation | 0.00 | 0.42 | 0.04 | 0.16 | 0.00 | 12.0 |
| FT* | 68.2 | 8.62 | 46.2 | 71.0 | 2.47 | 5.6 |
| Adapter[L](0.37M)* (Lin et al., 2020) | 66.3 | 8.41 | 45.0 | 69.8 | 2.40 | 9.6 |
| Adapter[L](11.09M)* (Lin et al., 2020) | 68.9 | 8.71 | 46.1 | 71.3 | 2.47 | 5.2 |
| Adapter[H]* (Houlsby et al., 2019) | 67.3 | 8.50 | 46.0 | 70.7 | 2.44 | 8.2 |
| FT[Top2]* (Li & Liang, 2021) | 68.1 | 8.59 | 46.0 | 70.8 | 2.41 | 7.8 |
| PreLayer* (Li & Liang, 2021) | 69.7 | 8.81 | 46.1 | 71.4 | 2.49 | 4.2 |
| LoRA (Hu et al., 2022) | 68.9 | 8.68 | **46.5** | 71.5 | 2.51 | 3.4 |
| LMNet | **70.5** | 8.85 | **46.5** | 71.5 | 2.48 | **2.2** |
| LMNet + FT | 66.0 | 8.46 | 42.4 | 68.3 | 2.05 | 10.2 |
| LMNet + Prefix | **70.5** | **8.88** | 46.2 | **72.4** | 2.46 | 2.4 |
| LMNet + LoRA | 70.1 | 8.82 | 46.2 | 71.7 | **2.54** | 2.4 |

To avoid over-fitting with LMNet, by default we only train the edge parameters and keep the vertex parameters frozen. We still construct LMNet with a small scale (3 layers with 1/2/1 vertexes), using an attention block, containing 5.25M parameters, for each edge module. Note that thanks to the fully-differentiable paths in LMNet, we can also integrate LMNet along with PEFT methods. For example, we implemented LMNet+Prefix (Li & Liang, 2021) by adding random-initialized and to be adapted prefix before the initial input $\mathbf{X}^{\text{in}}$. LMNet+FT and LMNet+LoRA mean not only updating edge parameters, but also updating vertex parameters in a fully-adaptation or low-rank adaptation manner respectively. The results are provided in Table 11. In terms of the average value of rankings under every evaluation metric, LMNet performs best, showing learning dense communication makes effective and generalizable adaptation. LMNet+Prefix wins under the most individual metrics, suggesting plugging-in appropriate PEFT methods can further improve the performance.

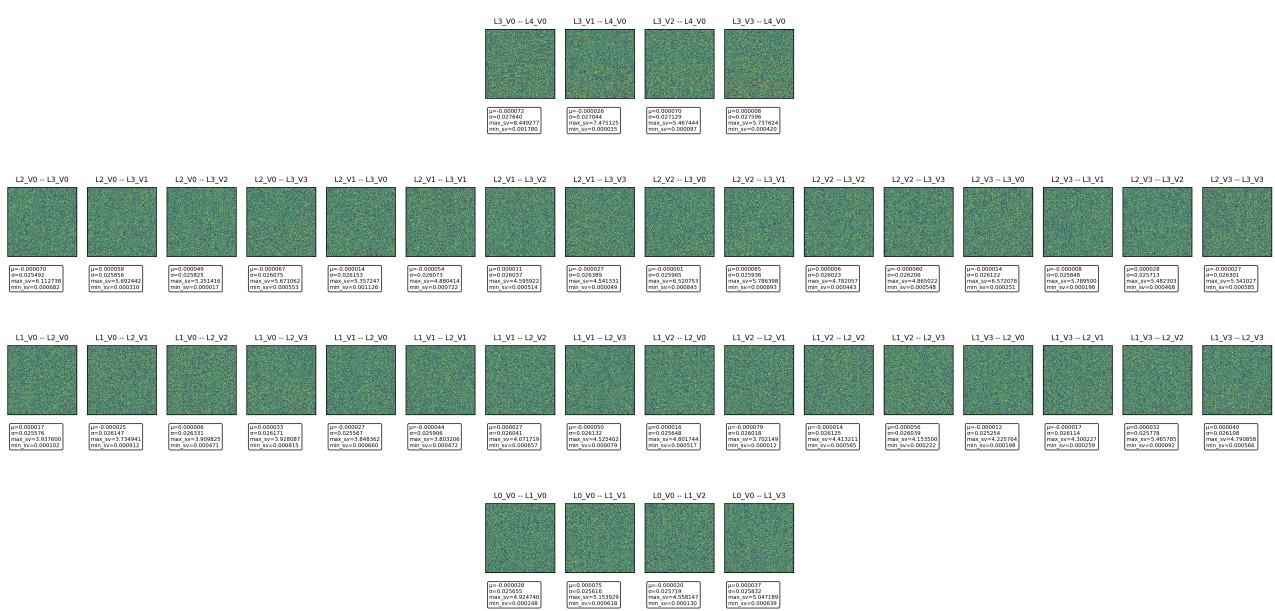

*Figure 4.* Visualization of query projection matrix of the attention block on every edge in trained LMNet. All edges are shown under the same value-color mapping.

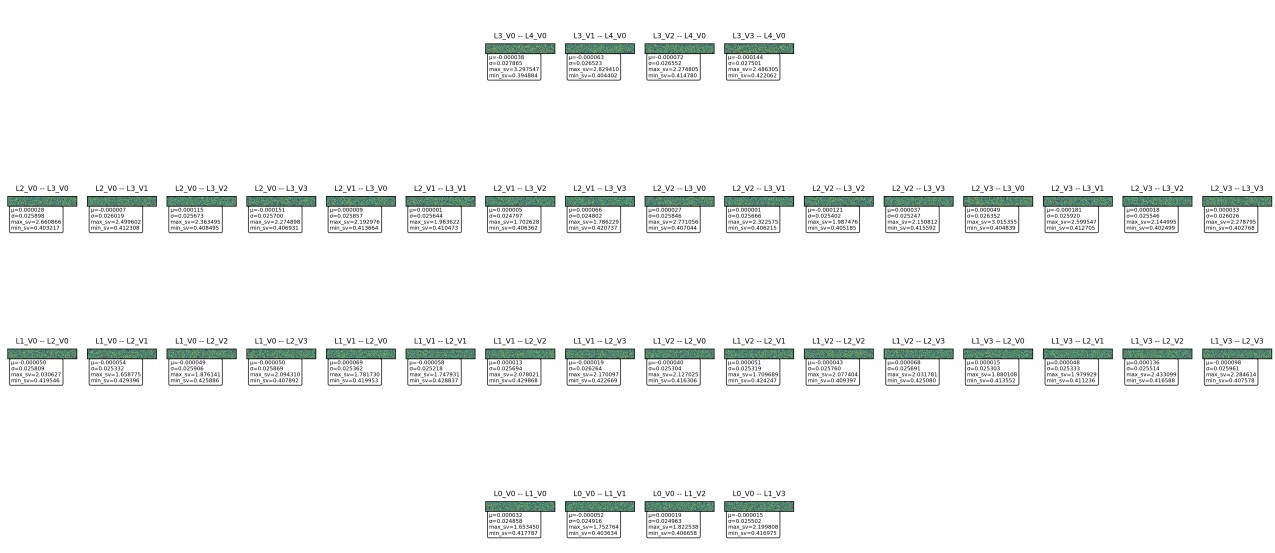

*Figure 5.* Visualization of key projection matrix of the attention block on every edge in trained LMNet. All edges are shown under the same value-color mapping.

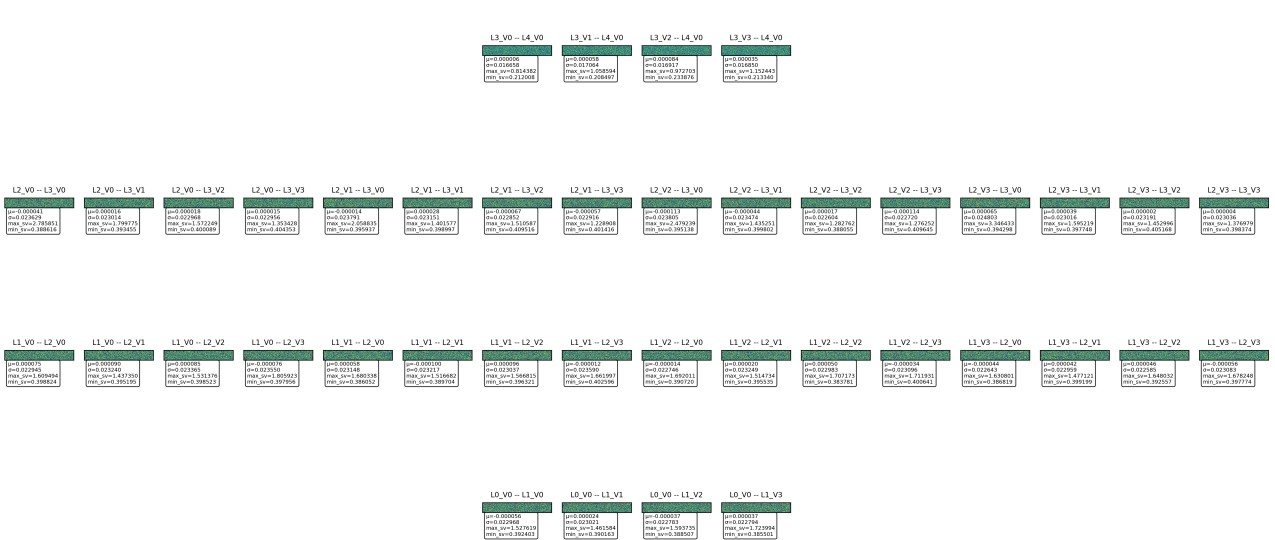

*Figure 6.* Visualization of value projection matrix of the attention block on every edge in trained LMNet. All edges are shown under the same value-color mapping.

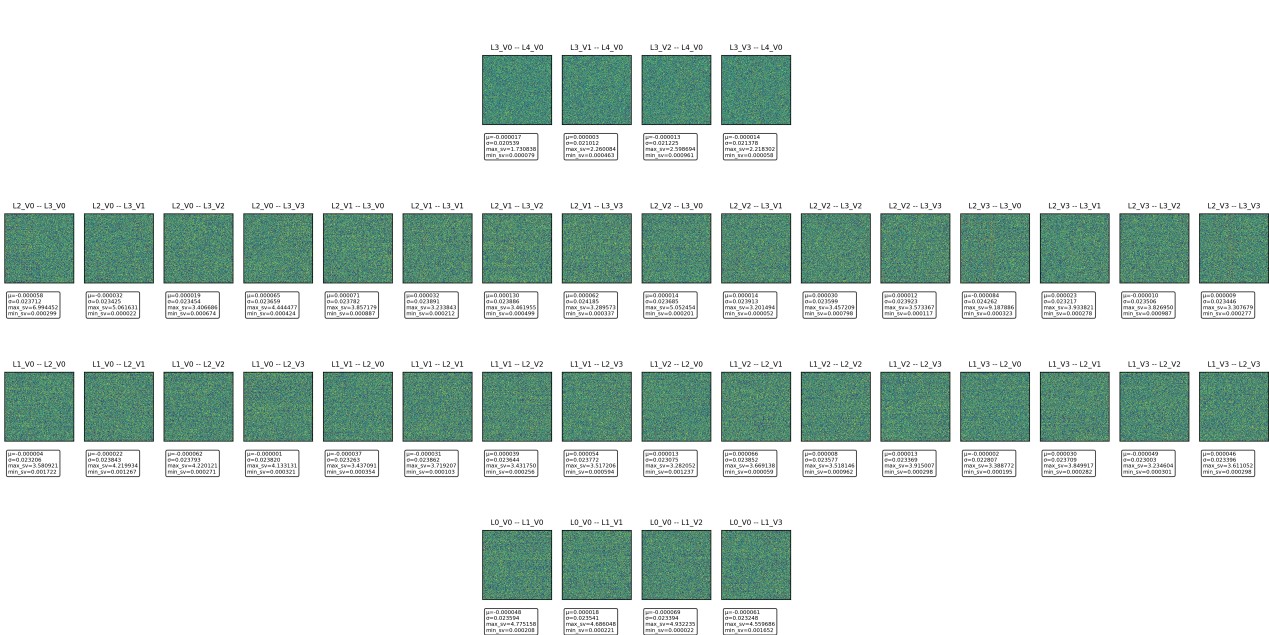

*Figure 7.* Visualization of output projection matrix of the attention block on every edge in trained LMNet. All edges are shown under the same value-color mapping.

