# OpenReview forum: "Language Model Networks: Supervision-Efficient Learning through Dense Communication"
_ICML.cc/2026/Conference — ICML 2026 regular_

### Official Review · Reviewer_2aXx · 2026-03-09

**Soundness:** 4
**Presentation:** 2
**Significance:** 4
**Originality:** 3
**Overall Recommendation:** 5
**Confidence:** 3

**Summary:**

In this work, the authors propose an abstraction of LLM inference by viewing them as nodes in a large network of interconnected LLM calls. They implement this by removing the embedding/unembedding components of each model and having them communicate in embedding space, connected via trained seq2seq models representing "edges" in the network. In practice the authors implement a graph structure resembling an MLP, with discrete tokens being embedded and passed into a "head" LM, then through an MLP-like architecture of LLMs, then to a singular decoder LLM with an unembedding matrix to map back to tokens. The authors demonstrate that this approach is better than simple test-time approaches (self-consistency and self-refinement) with the same number of models, better than SFTing a model for the same amount of time, and better than prompting iteratively. They also show that the approach is better than calling a larger model of similar total parameter size, and at much cheaper compute cost. Finally, they demonstrate how this approach is also more performant in terms of improving reasoning over similar latent-space reasoning methods and better than parameter-efficient fine-tuning for data-efficient adaptation.

**Compliance With Llm Reviewing Policy:**

Affirmed.

**Final Justification:**

The authors answered my questions, and I keep my review as an accept.

**Key Questions For Authors:**

* In 4.2.1, why does LMNet introduce $2.45\times10^8$ parameters, but 5.45M in 4.22 if they both use the 1-2-1 graph structure? Are they on different sized LLMs?

**Limitations:**

yes

**Strengths And Weaknesses:**

## Strengths
 * The idea is intuitive and clear
 * The results speak for themselves, demonstrating the efficacy of the method
 * The experiments do a good job of proof-of-concept while the authors admit their limited training budget. Studying performance efficiency against baselines, and the two applications for reasoning and data-efficient adaptation are great.

## Weaknesses
 * I think the appendix table (or just the one column) comparing with SFT'd models of the same parameter count as the full LMNet parameter count should be in the main paper. This is an important comparison as the current SFT column with 1 LLM from the LMNet architecture is slightly misrepresentative as the models are of different complexities.
 * In 4.2.1 and 4.2.2, are the LLMs frozen? Its a little unclear which times you are updating the LLM weights and which times you aren't. Sometimes you explicitly say it which confuses me when I encounter experiments where you don't mention it.
 * In the intro, you say "all these other methods are within this paradigm" 1) you haven't even described or introduced the paradigm yet, and 2) I would expect this claim to be substantiated more. Just because you are connecting LLMs together and other methods connect LLMs doesn't mean that they are within this paradigm. You could easily just describe the LMNet architecture for these methods (e.g., for Best of N sampling, its a width N network with identity edge weights and an argmax of some sort).

Overall, I think the contribution of this work is very high and the paper will be an important contribution to ICML. However, the writing has some typos and is a little sloppy at parts. With a good copy editor, this could be a top contribution to the conference.

## Typos:
 * 110: you repeat "Another line of work explores differentiable communication"
 * 234: It turns out that significantly outperform is incorrect grammar
 * Some others I can't remember where

---

> ### Author Rebuttal · Authors · 2026-03-31
>
> We sincerely thank the reviewer for the strong support and for the concrete suggestions on how to improve clarity and fairness.
>
> ### Q. “Why does 4.2.1 add many more parameters than 4.2.2 although both use a 1–2–1 graph?”
> Because the **topology is the same but the backbone model size is different**, and each edge is instantiated as a **backbone-matched transformer block**. Therefore, the parameter count of each edge scales with the hidden size and projection sizes of the corresponding backbone. Section 4.2.1 uses substantially larger modern backbones (e.g., Qwen2.5-1.5B), whereas Section 4.2.2 uses GPT2-M (following LoRA), so the added parameter counts differ markedly despite identical topology. We will clarify this explicitly in the paper.
>
> ### W1. “The same-size comparison should be in the main paper.”
> We agree that this comparison is important for calibration. This is exactly the role of Appendix B / Table 7, which compares LMNet-1B with monolithic pre-trained LLMs of similar size. At the same time, one reason we did **not** place it in the main paper is that it is still not a strictly apples-to-apples comparison: although the model sizes are similar, the **pre-training cost/data/compute** behind these monolithic models differ enormously, whereas LMNet-1B is built by reusing an existing pre-trained backbone with modest additional training.
>
> That said, we agree that readers should see this comparison more clearly. In a revision, we will reconsider its placement and framing so that its role is explicit: useful for contextualization, but not a perfectly matched training-investment comparison. If the reviewer’s suggestion is instead to include a **matched monolithic SFT baseline at the full LMNet parameter count**, we agree that this would also be a valuable control; we do not currently include that exact experiment and will avoid implying otherwise.
>
> ### W2. “It is unclear in Sections 4.2.1 and 4.2.2 when the LLM weights are frozen.”
> Thank you; we agree this should be much clearer. The training protocols are:
>
> - **Section 4.1:** first freeze vertices and train edges; then jointly train all LMNet parameters.
> - **Section 4.2.1 (MMLU/GSM8K):** jointly update **all** LMNet parameters.
> - **Section 4.2.2 (E2E):** freeze all vertex LLM parameters and update **only** the edge parameters.
>
> We will make this explicit at the beginning of each experiment subsection and summarize it in a compact table.
>
> ### W3. “The introduction overstates that prior methods are ‘within this paradigm.’”
> We agree. The wording is too broad. What we intended to say is that many existing methods can be **viewed through the lens of communication topology and information flow among LLM computations**, not that they literally instantiate the LMNet paradigm. We will revise the introduction accordingly.
>
> ### “Writing has typos / grammar issues.”
> We thank the reviewer for the careful review and appreciate the specific examples. We will fix the cited issues and perform a thorough proofreading pass.

---

> > ### Author Rebuttal · Reviewer_2aXx · 2026-04-01
> >
> > The authors answered my questions!

---

### Official Review · Reviewer_BM1W · 2026-03-12

**Soundness:** 2
**Presentation:** 3
**Significance:** 2
**Originality:** 2
**Overall Recommendation:** 4
**Confidence:** 4

**Summary:**

The paper proposes LMNet, a high-level neural network architecture that treats pre-trained LLMs as optimizable nodes connected via trainable sequence-to-sequence modules (edges). To bypass the informational bottleneck and optimization limitations of discrete natural language, LMNet strips the embedding and de-embedding layers from its intermediate models, allowing them to communicate directly through continuous dense vectors. Configured in a layer-wise, fully connected graph similar to an MLP, the entire multi-model system is end-to-end differentiable and can be optimized via gradient descent. The authors demonstrate that this approach improves general intelligence and facilitates data-efficient adaptation compared to standard prompting and fine-tuning techniques.

**Compliance With Llm Reviewing Policy:**

Affirmed.

**Final Justification:**

All my concerns have been addressed.

**Key Questions For Authors:**

1. Could the authors compare LMnet with the methods mentioned in "Cons"?

**Limitations:**

No. The authors may discuss the safety and alignment risks associated with abandoning discrete natural language communication. Because LMNet's intermediate continuous vectors cannot be reliably decoded into coherent text , researchers completely lose the ability to audit the network's reasoning process.

**Strengths And Weaknesses:**

Pros:
1. By shifting from discrete token communication (≈17 bits/token) to continuous dense vector communication (≈28,672 bits/vector), the architecture preserves the rich, high-dimensional uncertainty and nuance of the model's internal states.
2. Abandoning non-differentiable text generation for intermediate steps allows for continuous gradient flow across multiple LLM agents, enabling joint optimization of the entire system.
3. By sharing weights across the vertex LLMs, the framework provides a relatively scalable way to construct a deep, multi-agent reasoning network without multiplying the massive core parameter count of the foundational models.

Cons:
1. The use of full attention blocks as edge modules introduces massive parameter overhead (e.g., 600 million parameters just for communication). The authors fail to justify this against zero-parameter continuous communication alternatives like Activation Grafting, which achieve similar results using simple mathematical pooling.
2. LMNet relies solely on the final autoregressive output loss. It lacks explicit regularizers (such as conditional separation losses found in contemporary latent frameworks) to ensure intermediate vectors do not devolve into uninterpretable statistical noise.
3. Distributing computation across a massive spatial graph incurs high latency and memory costs. The paper ignores intra-model temporal routing techniques (like TurboConn) that achieve similar continuous reasoning depth within a single model at zero additional memory cost.
4. The authors concede that the intermediate continuous vectors cannot be decoded into coherent text, meaning the system's internal reasoning process is entirely opaque to human auditing.
5. in the general intelligence experiment, an LMNet built from a 0.5B parameter model (with 1.1B total parameters and ~14x the computation per token) is compared to the base 0.5B model and its test-time scaling variants. While LMNet performs better, it is difficult to conclude that this is due to the superiority of the architecture itself, rather than the massive increase in parameters and compute. The appendix comparison to monolithic models of similar size shows a more modest, comparable performance, which tempers the main claim. The experiments on customizing LLMs with limited data are better controlled and more convincing, showing clear benefits over PEFT methods and other latent reasoning approaches. The analysis of the learned communication is interesting but remains somewhat speculative.

---

> ### Author Rebuttal · Authors · 2026-03-31
>
> We thank the reviewer for the thoughtful review. We will place the mentioned methods as important related works in the revision of manuscript.
>
> We additionally compare the following baselines under Sec. 4.2.1 with Qwen2.5-1.5B:
> - **LMNet**: proposed.
> - **LMNet(0 param)**: the same 1–2–1 graph, but remove trainable edge modules and directly average predecessor hidden states.
> - **LMNet(linear)**: the same 1–2–1 graph, but replace each transformer edge by a token-wise linear projection.
> - **Activation Grafting (AG)**: run two models with slightly different prompts, and fuse and replace them at layer 24 using a learned `3072 x 1536` linear projection.
> - **TurboConn (TC)**: training 20 low-rank linear connectors (`1536 x 120 x 2` each) connecting layers `9..28` to `1..20`.
>
> |  | MMLU | GSM8K |
> | --- | ---: | ---: |
> | LMNet | **37.3** | **72.7** |
> | LMNet(0 param) | 34.2 | 55.2 |
> | LMNet(linear) | 35.1 | 62.8 |
> | Activation Grafting | 34.5 | 59.6 |
> | TurboConn | 35.3 | 65.0 |
>
> ### Q: “Compare with Activation Grafting / TurboConn”
> We have now added the above comparison. AG and TC are relevant as they can also be viewed as communicate through hidden states, but they address a different architectural question. AG and TC focus on **intra-model, token-level** routing across layers within one model, whereas LMNet studies **inter-node, sequence-level** communication in a trainable graph with branched topology.
>
> We will revise the paper to position these methods more clearly in the related work and discussion. At the same time, we would prefer not to overstate the empirical comparison in the manuscript: to the best of our knowledge, AG and TC do not have public implementations, so this is not canonical reproductions of the original methods. Moreover, they are directly plug-compatible in our broader setting (4.1 and 4.2.2). and TC is also a such recent contemporary work (Feb,26) that is impossible to notice in this submission (Jan,26). We therefore view the current comparison as informative but not definitive.
>
> ### W1: “The edge modules are heavy; Activation Grafting.”
> Please note that AG is NOT parameter-free in practice, since it requires a learned projection. We also note that the current edge module is not individually heavy: in the 1/4/4/4/1 LMNet, the reported 0.6B edge parameters are the sum over 41 edges, so each edge module has only about 14M parameters.
>
> The ablation above suggests that simply passing hidden states, removing sequence modeling from the edgedoes not recover the full gain of LMNet. This supports the view that the seq2seq edge module provides a meaningful advantage over token-level alternatives.
>
> ### W2&4: “only final autoregressive loss and lacks explicit regularizers to prevent intermediate vectors from becoming uninterpretable noise. The internal reasoning process is entirely opaque to human auditing.”
> The paper aims to answer **whether useful latent communication can emerge under end-to-end task supervision?** Our evidence in Sec 5 is still preliminary but suggests yes: some intermediate vertex outputs remain partially decodable and meaningful, rather than collapsing into arbitrary noise.
>
> Meanwhile, we would soften the statement that the process is "entirely opaque": the intermediate communication is not entirely opaque (see the case study), but it is clearly harder to inspect than natural-language communication, and the current vectors cannot be reliably decoded into coherent text. We agree that auditability matters and will make this limitation more explicit in the revision.
>
>
> ### W3: “The spatial graph incurs substantial latency/memory cost, and the paper ignores intra-model temporal routing techniques such as TurboConn.”
> Please note that TC is NOT zero-memory either, since it introduces dozens of connector projections. We agree that LMNet is more expensive than a single-backbone baseline, and this needs to be framed carefully, as discussed in Sections 3.3 and 7.
>
> Please refer to the AW4X response for the latency comparison. Our claim is not that LMNet is a free efficiency gain; rather, it offers a different trade-off among performance, latency, and memory.
>
> ### W5: “the gain may come from extra parameters/compute rather than architecture.”
> The gain does come from extra parameters/compute, **but only under reasonable architecture design**.
> The newly added experiment suggests that the gain is not explained by simply adding any latent connector or lightweight routing mechanism: LMNet still outperforms the alternatives above.
> The paper is not trying to argue that LMNet universally outperforms any monolithic model of similar size. Rather, it proposes a method for improving a given backbone through learned latent communication, with relatively small additional training compared with pre-training a new monolithic model from scratch.

---

> > ### Author Rebuttal · Reviewer_BM1W · 2026-04-03
> >
> > I'd like to thank the authors for addressing all my concerns. I will raise the score.

---

### Official Review · Reviewer_AW4X · 2026-03-13

**Soundness:** 3
**Presentation:** 3
**Significance:** 3
**Originality:** 3
**Overall Recommendation:** 4
**Confidence:** 3

**Summary:**

This paper introduces LMNet, a framework that treats pre-trained LLMs as nodes in a higher-level neural network, connecting them via continuous dense vectors rather than discrete natural language tokens. The authors strip the embedding and de-embedding layers from intermediate LLMs, insert trainable seq2seq edge modules (single attention blocks) between them, and arrange everything in a layer-wise fully connected topology analogous to an MLP. The whole system is end-to-end differentiable and trained with autoregressive loss. Two applications are demonstrated: (1) improving general intelligence by building a 5-layer LMNet from shared Qwen2.5-0.5B vertices, showing large gains on benchmarks like MMLU, BBH, and MATH compared to prompting and test-time scaling baselines; and (2) customizing LLMs with limited data, where LMNet competes favorably with PEFT methods like LoRA on the E2E dataset and outperforms latent reasoning baselines (COCONUT, CODI) on MMLU and GSM8K. The paper also provides micro- and macro-level analyses of the learned communication patterns.

**Compliance With Llm Reviewing Policy:**

Affirmed.

**Key Questions For Authors:**

Have you measured actual wall-clock inference time for LMNet-1B versus simply running Qwen2.5-1.5B (or even 3B) on the same benchmarks? Given that LMNet requires 5 sequential passes through the vertex model plus edge module overhead, I suspect the latency is substantially worse than a single larger model that achieves similar or better accuracy

**Limitations:**

yes

**Strengths And Weaknesses:**

Strengths:
1. the paper is well written and easy to follow
2. The core idea of bypassing discrete token bottlenecks between LLMs is clean, well-motivated by an intuitive information-theoretic argument, and the formalization as a directed graph with stripped transformers and trainable edges is elegant.

Weakness:
1. My main concern is the experiments are limited to very small models and the scalability story is thin. Nearly all results use 0.5B–1.5B parameter models, with only a brief ablation going up to 7B in the limited-data setting rather than the more impressive general intelligence setting. The paper's motivating narrative is not backed by evidence at meaningful scale. The gains could shrink or vanish with stronger base models, and the GSM8K results with Qwen2.5-1.5B already hint at diminishing returns for easier tasks.

2. Latency scales roughly as L × (vertex forward pass), meaning 5× for the general intelligence LMNet, but there's no wall-clock timing, no throughput comparison, and no latency-vs-accuracy Pareto analysis. LMNet-1B with 5 sequential passes of a 0.5B model may well be slower than just running Qwen2.5-1.5B, which often outperforms it.

3. The authors themselves note the end-to-end autoregressive implementation doesn't match the "sentence-by-sentence communication" picture that motivates the paper, the macro-level analysis just confirms edges are non-trivially trained without revealing what structure emerged, and there's no rigorous probing (e.g., ablating edges, measuring information flow) to substantiate the "Society of Mind" framing beyond aspiration.

---

> ### Author Rebuttal · Authors · 2026-03-31
>
> We thank the reviewer for the positive assessment of the paper’s motivation, formulation, and presentation.
>
> ### Q: "Have you measured actual wall-clock inference time for LMNet-1B versus simply running Qwen2.5-1.5B (or even 3B)?"
> We appreciate this important question. The current submission reports a complexity discussion in Section 3.3; to address the reviewer’s concern more directly, we additionally measured preliminary **wall-clock end-to-end inference time** against **Qwen2.5-1.5B**.
>
> Our intended point is narrower than a deployment claim. With vertex parameter sharing, the **critical-path per-token latency** scales mainly with the number of LMNet layers, because vertices in the same layer can be executed in parallel and the edge modules are much smaller than the vertex LLMs. At the same time, **end-to-end response time** also depends on the number of generated tokens. In practice, LMNet often produces shorter final outputs than the monolithic baseline, so the measured wall-clock increase is materially smaller than a naive depth-based extrapolation would suggest.
>
> Concretely, comparing LMNet-1B with Qwen2.5-0.5B/1.5B/3B, we have:
>
> **Accuracy**
>
> | Dataset | Qwen2.5-0.5B | Qwen2.5-1.5B | Qwen2.5-3B | LMNet-1B |
> | --- | ---: | ---: | ---: | ---: |
> | BBH | 20.3 | 45.1 | 56.3 | 47.3 |
> | GSM8K | 41.6 | 68.5 | 74.8 | 50.3 |
> | GPQA | 24.8 | 24.2 | 25.6 | 25.6 |
>
> **Output Tokens**
>
> | Dataset | Qwen2.5-0.5B | Qwen2.5-1.5B | Qwen2.5-3B | LMNet-1B |
> | --- | ---: | ---: | ---: | ---: |
> | BBH | 210.4 | 195.2 | 204.4 | 140.6 |
> | GSM8K | 281.9 | 243.5 | 260.7 | 158.8 |
> | GPQA | 198.4 | 256.2 | 303.0 | 189.1 |
>
> **Time (seconds)**
>
> | Dataset | Qwen2.5-0.5B | Qwen2.5-1.5B | Qwen2.5-3B | LMNet-1B |
> | --- | ---: | ---: | ---: | ---: |
> | BBH | 1.92 | 3.48 | 6.62 | 4.99 |
> | GSM8K | 2.44 | 3.91 | 7.76 | 5.08 |
> | GPQA | 2.01 | 3.81 | 8.89 | 5.62 |
>
> Take the comparison with model with most similar size, Qwen2.5-1.5B, for example.
> Across these three benchmarks, LMNet's latency **per generated token** is about **2×**, while **end-to-end runtime** increases by only **1.30×–1.48×**, because LMNet uses substantially fewer output tokens. We therefore want to calibrate the claim carefully: LMNet is **slower at inference** than a single monolithic model, but the wall-clock gap is smaller than the layer count alone would suggest.
>
> ### W1. “Experiments are limited to very small models; the scalability story is thin.”
> We agree that this is the main limitation of the current submission. Our central claim is **methodological**: the paper provides initial evidence that graph-structured latent communication among LLM nodes is feasible and can be beneficial at small-to-medium scale.
>
> We would like to clarify two supporting observations already present in the submission. First, in the **limited-data setting**, we include ablations on **3B and 7B** vertex models (Table 5), where LMNet continues to show gains. Second, in Appendix D, the gains are **larger on harder tasks** than on easier ones; in particular, the improvement is more substantial on the harder OpenR1-Math subset than on GSM8K. That said, we agree this does **not** substitute for a larger-scale experiment in the main “general intelligence” setting, and we will revise the paper to frame the current evidence as a proof-of-concept rather than a strong scalability demonstration.
>
> ### W2. “No wall-clock timing / no latency-vs-accuracy Pareto analysis.”
> This is exactly the issue addressed above. We agree that explicit timing is more informative than asymptotic discussion alone. In a future version, we will include broader inference-time comparisons, including stronger monolithic baselines and a fuller latency–accuracy analysis.
>
> ### W3. “The current analysis does not fully substantiate the ‘Society of Mind’ framing.”
> We appreciate this point and agree that the current analysis is **suggestive rather than definitive**. In particular, Appendix A.3 already notes that the current end-to-end auto-regressive implementation does not exactly match the motivating image of sentence-by-sentence communication, and the present macro/micro analyses are not sufficient to establish a strong mechanistic claim.
>
> We will therefore revise the paper in two ways. First, we will **tone down** the “Society of Mind” framing from an empirical claim to a motivational interpretation. Second, we will clarify that more rigorous probing—e.g., edge ablations, causal intervention, or information-flow analysis—is needed before making stronger claims about the learned mechanism.

---

> > ### Author Rebuttal · Reviewer_AW4X · 2026-04-03
> >
> > Thanks for your rebuttal. I will maintain my original score.

---

### Decision · Program_Chairs · 2026-04-30

**Decision:**

Accept (regular)

**Comment:**

This paper proposes LMNet, a framework that treats entire LLMs as nodes in a high-level neural network. Mechanically, this involves learning a seq2seq module that communicates between LLMs. The entire network remains differentiable and can be trained end-to-end. Experiments demonstrate the promise of this approach on a variety of benchmarks.

All reviewers agreed that the method was interesting and novel, and backed up by solid performance. There were some concerns about (1) increased wallclock speed, and (2) the fact that experiments were only done with 0.5B models. However, given that this is a methodological contribution, I don't think these are too big of concerns.

This is a solid contribution to the conference.